



# Meteor echo height ceiling effect and the mesospheric temperature estimation from meteor radar observation

Changsup Lee[1], Geonhwa Jee[1], Jeong-Han Kim[1], In-Sun Song[1]

[1]Polar Climate Sciences Research, Korea Polar Research Institute, Incheon, 21990, South Korea

*Correspondence to*: Changsup Lee (cslee@kopri.re.kr)

**Abstract.** The mesospheric temperature estimation from meteor height distribution is reevaluated by using the Sounding of the Atmosphere using Broadband Emission Radiometry (SABER) and the King Sejong Station meteor radar observations. It is found that the experimentally determined proportionality constant between the full width at half maximum (FWHM) of the meteor height distribution and temperature is in remarkable agreement with theoretical value derived from the physics-

based equation and it is nearly time-invariant for the entire observation period of 2012-2016. Furthermore, we newly found that the FWHM provides the best estimate of temperature at slightly lower height than the meteor peak height (MPH) by about 2-3 km. This is related to the asymmetric distribution of meteor echoes around MPH, which is known to be caused by the meteor echo height ceiling effect (MHC). At higher altitude above MPH, the meteor detection rate is greatly reduced due to the MHC and the cutoff height for this reduction follows a fixed molecular mean free path of the background atmosphere.

This result indicates that the meteor height distribution can be used to estimate the mesospheric temperature even under the asymmetric meteor echo distribution caused by the MHC at high altitude.

## 1 Introduction

Recent advances in the performance of meteor radar have enabled continuous observations for the daily mesopause temperature and hourly neutral winds in the mesosphere and lower thermosphere region. As meteoroids enter the earth's

atmosphere, they undergo ablation due to collisional heating with atmospheric constituents, leaving cylindrical ionized meteor trails behind them. By observing these meteor trails with a meteor radar, one can extract a variety of essential information on the background atmosphere as well as the meteors. While the neutral winds can be directly obtained from the measurement of Doppler shift of backscattered signals, the mesopause temperature has been conventionally estimated from the diffusion coefficients of underdense meteor echoes based on the dependence of the diffusion coefficient on the

atmospheric temperature and pressure (Chilson et al., 1996; Kim et al., 2012, and references therein). However, Holdsworth et al. (2006) suggested that the width of the meteor height distribution can be used for an alternative temperature estimation procedure. Lee et al. (2016) demonstrated that there exists a clear linear relationship between the full width at half maximum (FWHM) of the height distribution of detected meteor echoes and the temperature retrieved from the Aura Microwave Limb Sounder (MLS) based on a basic theory and observations. They further showed that the temperature estimated from this





relation is in better agreement with satellite temperature measurements compared with conventionally estimated temperature from meteor decay times. Although it was successfully shown that meteor height distribution provides mesospheric temperature, the MLS temperature data has a poor height resolution (~10 km), which is nearly comparable to the FWHM in the mesosphere. Therefore, the resulting temperature from the FWHM was assumed to be a layer mean temperature at near

the meteor peak height (MPH). Furthermore, a meteor radar has a limitation on the height range of meteor detection; it depends on radar specifications such as a pulse repetition frequency and a radio wavelength.

In this study, we reexamine the temperature estimation procedure from the FWHM with the emphasis of the invariance of proportionality constant between the FWHM and background temperature not only from theoretical consideration but also

from meteor radar and TIMED/SABER observations. In addition, we also evaluate the validity of temperature estimation from the FWHM under the meteor echo height ceiling effect (MHC). Section 2 describes a theoretical derivation of the linear relationship between the FWHM and background temperature. The results of this study are presented in section 3 with relevant discussions. Finally, this is followed by a conclusion in section 4.

## 2 Observations

### 2.1 King Sejong Meteor Radar

Meteor radar has been used to continuously monitor atmospheric winds and temperatures in the mesosphere and lower thermosphere for several decades. Korea Polar Research Institute (KOPRI) has been operated a meteor radar at King Sejong Station (KSS) in Antarctica (62.22°S, 58.78°W) in collaboration with Chungnam National University, Korea, since March 2007. The KSS meteor radar using a frequency of 33.2 MHz transmits 7.2 km width, 4-bit complimentary coded circularly

polarized pulses at a pulse repetition frequency of 440 Hz. The transmitter has a peak power of 12 kW and a duty cycle of 8.4%. The receiver is composed of two perpendicular interferometric baselines to determine the angle of arrival of backscattered signal from meteor trails (Lee et al., 2013).
It collects underdense meteor echoes within a horizontal radius of about 250 km from the radar site. The number of meteor echoes from the KSS meteor radar reaches up to 40,000 meteors per day in summer but it declines to about 15,000 in winter.

The large number of meteor echoes enables us to obtain reliable meteor samples even beyond the typical meteor detection height of 80-100 km with a better temporal resolution.

In this study we used 5-year-long meteor radar data from 2012 to 2016 to ensure better statistics of meteor distribution even under the minimized meteor detection rate in winter. Phase error of meteor echoes is limited to be less than 6-degree to

determine the most accurate meteor height distribution. In deriving a linear relationship between the width of meteor height



distribution and the SABER temperature, the geometric height of meteor echoes was converted to geopotential height to correctly compare with the proportionality constant derived from the fundamental hydrostatic equation.

## 2.2 TIMED/SABER

The Sounding of the Atmosphere using Broadband Emission Radiometry (SABER) instrument is one of four instruments on
NASA's TIMED (Thermosphere Ionosphere Mesosphere Energetics Dynamics) satellite to measure the limb emission in the ten broadband infrared channels covering from 1.27 $\mu$m to 17 $\mu$m. The profile of kinetic temperature is obtained from the 15 $\mu$m radiation of $CO_2$ from 15 km to 120 km altitude.

The SABER instrument views the atmospheric limb perpendicular to the satellite orbital track in an altitude of about 625 km and an inclination of 74°. In order to keep the SABER instrument on the anti-sunward side, the TIMED satellite performs
yaw maneuvers about every 60-day period. Consequently, the latitude coverage on a given day extends from about 52° in one hemisphere to 83° in the other and this results in only six months of SABER data available every year in high latitude regions above 52°. The height resolution of the data varies with altitude and it is about 2 km in the region of meteor detection. The SABER data used in this study are version 2.0, which includes non-LTE temperature inversions in the upper mesosphere and lower-thermosphere due to the departure from LTE in the $CO_2$ 15 $\mu$m vibration-rotation band for the kinetic temperature
determination above 70 km altitude (Mertens et al., 2001; 2004). The SABER temperature and geopotential height data were restricted to the distance of less than 500 km from the location of KSS to directly compare with the FWHM derived from meteor radar observations during the period of 2012-2016.

## 3 Theoretical considerations of FWHM and temperature

According to Lee et al., (2016), most of the observed underdense meteor echoes show specific height distributions being
primarily determined by background atmospheric pressure. Figure 1 shows the MPH (blue open squares) and FWHM (red-shaded area) obtained from the fitting procedure with a Gaussian curve applied to the daily meteor height distribution from 2012 to 2016. The background atmospheric pressure field from the MLS measurement is also presented by solid line contours. It is important to note that the MPH closely follows the constant pressure level and a fixed portion of the height distribution (i.e., FWHM) of observed meteor echoes exists within two constant pressure levels around the MPH as shown in
Figure 1. As meteors penetrate into the Earth's atmosphere down to about 120 km height, they produce meteor trails, which are composed of metallic ions and electrons by collisions with atmospheric constituents. This collisional heating process is critically affected by background atmospheric pressure which is a function of density and temperature. Therefore, the height distribution of meteor echoes, represented by the FWHM, is determined by the state of the background atmosphere.

The linear relationship between the FWHM and temperature can be derived from the conventional atmospheric statics: the variation of pressure with height can be determined from the ideal gas law and the hydrostatic equation (Andrew et al., 1987):





$$\frac{\partial \ln P}{\partial z} = -\frac{g}{RT},$$ (1)

where g and R are the gravitational acceleration and gas constant, respectively. After a simple rearrangement for separation

of variables, both sides in the Eq. (1) can be integrated over the region between two given constant pressure levels of $P_1(Z_1)$ and $P_2(Z_2)$ to obtain the hypsometric equation:

$$Z_2 \text{-} Z_1 = \frac{R}{g} \int_{P_2}^{P_1} T d \ln P.$$ (2)

The height difference $Z_2\text{-}Z_1$ in Eq. (2) corresponds to an atmospheric layer between the two constant pressure levels. Since the FWHM of the meteor height distribution nearly coincides with the atmospheric layer as in Figure 1, it can be used to estimate the mean temperature of the layer from the Eq. (2):

$$\langle T \rangle = C \cdot \text{FWHM},$$ (3)

where $\text{FWHM} = Z_2\text{-}Z_1$ and the proportionality constant $C = \frac{g}{R}\left[\ln\left(\frac{P_1}{P_2}\right)\right]^{-1}$. Here the layer mean temperature is defined as:

$$\langle T \rangle = \frac{\int_{P_2}^{P_1} T d \ln P}{\int_{P_2}^{P_1} d \ln P}.$$ (4)

As is revealed from the definition of the layer mean temperature given by Eq. (4), the mean temperature can be defined for any kinds of temperature profiles even vertically rapidly varying temperature structure in atmosphere.

Eq. (3) clearly shows that the neutral temperature near the meteor peak height can be determined by FWHM alone with a proportionality constant. The constant can empirically be determined based on a linear relationship between the observed FWHM and temperature. It turns out that the determined proportionality constant dose not vary with time and can be

considered to be a 'constant' over the entire observation period. The constant can also be estimated with pressure measurements from SABER observations. The ratio between two pressure levels, $P_1/P_2$ is determined to be 7.59 from the SABER pressure measurements during the period of 2012-2016. Then the proportionality constant in Eq. (3) can be estimated to be about 16.28 when the gravitational constant $g$ and gas constant $R$ are approximately 9.47 and 287.06, respectively, in the region of given pressure levels of $P_1$ and $P_2$ near 90 km altitude. In the following section, we will

empirically determine the constant using the measurements of FWHM and temperature and will compare it with the estimated constant from the pressure measurements.





## 3 Results and Discussions

### 3.1 Empirical estimation of proportionality constant

Using the FWHM and temperature measured from the KSS meteor radar and SABER, respectively, we can determine the proportionality constant during 2012-2016 period. Figure 2 shows the scatter plots of the daily FWHMs derived from the

KSS meteor radar versus the $T_{SABER}$ at around 87 km for a year of 2013 (a) and for the entire observation period of 2012-2016 (b). In contrast to MLS temperature data used in Lee et al. (2016), SABER temperature measurements above KSS are only available in its south viewing geometry due to yaw maneuvers about every 60 days. This observational limitation gives rise to fewer temperature data points available for the determination of proportionality constant, which is why there are few data points in the middle of the scatterplot in Figure 2. Nevertheless, it has a much better height resolution than MLS

temperature measurement: the height resolution of SABER observation is about 2 km while the resolution of MLS observation is about 10~13 km, which is almost comparable to the FWHM. This characteristic of SABER observation allows us to find the representative altitude of the estimated temperature.

There is an obvious linear relationship between $T_{SABER}$ and FWHM with notably high correlation coefficients. The slopes in

Figure 2 represent the proportionality constant between the FWHM and $T_{SABER}$. Table 1 shows yearly slopes during the 5-year observation period. Note that the slopes are almost invariable within the associated error ranges during the entire observation period of 2012-2016. They also agree well with the theoretical values of the proportionality constant in Eq. (3) with SABER pressure measurements. Lee et al. (2016) using the Aura/MLS temperature data obtained notably smaller slope value of 15.71 with a worse correlation coefficient between the FWHM and temperature, which might be due to the poor

height resolution of MLS temperature data in the MLT region. It should be emphasized that the essential point of this procedure is the invariance of the proportionality constant between the FWHM and temperature near the MPH. Therefore, once it is determined from the independent measurement of temperature, it can be used to estimate the temperature from the meteor radar observation of the FWHM alone without any additional assumed parameter.

### 3.2 Meteor echo height ceiling effect on the temperature estimation

The estimated temperature using Eqs.(2)-(4) is the mean temperature between the two constant pressure levels as shown in Figure 1 and then it seems plausible that the mean temperature represents the temperature at around the meteor peak height (MPH) for the pressure levels around the FWHM. In order to confirm this representative altitude of the estimated temperature with the FWHM, we performed a correlation analysis between the FWHM and layer mean temperatures at different altitudes. Figure 3 shows the height profiles of the correlation coefficient between the FWHM and SABER

temperature during the period of 2012-2016. For this analysis the SABER temperatures were averaged at every 1.2 km height within 2.4 km width to obtain daily layer-mean temperatures for each year. It is clear in the figure that the best correlation occurs at slightly lower height (~86 km) than the MPH (88-91 km) by about 3-4 km. The temperature estimation





procedure using the meteor decay times, however, assumed that the representative altitude of the estimated temperature is around the meteor peak height, which is about 90 km altitude (Kim et al., 2012; Meek et al., 2013). A notable asymmetry in the correlation coefficients around the maximum correlation height is another important feature in Figure 3. The correlation coefficient more rapidly decreases at the altitude above the MPH than below and this indicates that the meteor height

distribution above the MPH is not only controlled by the background atmospheric state but other factors must be also involved.

The height distribution of meteor echoes detected by meteor radar depends not only on the physical characteristics of meteors and the state of the atmosphere but also on the operational parameters of meteor radar such as a radio wavelength

and a pulse repetition frequency. Meteor radar observation shows limited height range of detecting meteors for a given radio wavelength. The backscattered signals from meteor trails beyond this range are significantly attenuated to be detected. This limitation is inherently present in the meteor radar observations, which is known as the meteor echo height ceiling effect (MHC). Immediately after meteor ionized trails are formed, they rapidly expand in a radial direction to reach a finite radial extent called an initial radius within the interval that meteoric ions are in thermal equilibrium with surrounding atmosphere.

As the atmospheric density decreases with increasing height, the initial radius of meteor trail gets increased and becomes greater than a quarter of the radio wavelength, which significantly attenuates echo strength due to the lack of phase coherence from the signals reflected from the different spots in the meteor trail cross-section. In general, the meteor trails from fast meteors are produced at higher altitudes and hence meteor radar observation misses the significant part of meteors above certain altitude because of the MHC (McKinley, 1961; Campbell-Brown and Jones, 2003).


According to the echo attenuation theory, there are three major factors controlling the attenuation in the amplitude of meteor echoes from underdense meteor trails. Previous studies reviewed these attenuation factors and quantified their influences on MHC (Thomas et al., 1988; Steel and Elford, 1991). Since the detailed examination of three attenuation factors is beyond the scope of this study, we only give a brief overview of them and find which one is the most important in meteor echo

attenuation. The reduced electron density in the meteor trail with a larger initial radius makes backscattered signal too weak to be detected by radars (Initial radius factor, $\alpha_r$). The signal attenuation is also generated by the diffusion during the time of meteor trail formation due to the finite velocity of the meteoroid (Finite velocity factor, $\alpha_V$). If the inter-pulse period of a meteor radar is comparable or longer than the meteor decay times, it is more likely that meteor trail detected by one pulse decays below the threshold of meteor recognition before the arrival of successive pulse (Pulse repetition rate factor, $\alpha_P$).


In the temperature estimation procedure using the FWHM of meteor height distribution, it is critically important to take account of MHC caused by these attenuation factors on the meteor radar observations to more precisely determine the FWHM and, in turn to obtain atmospheric temperature with a minimized uncertainty. In this study, we calculated the three



attenuation coefficients using key parameters obtained from meteor radar observations to examine how much the FWHM can be affected by MHC and how it can influence on the temperature estimation.

We applied an attenuation theory described in Steel and Elford (1991) and Ceplecha et al. (1998) to the KSS meteor radar

data to calculate the attenuation coefficients. Figure 4 presents the height profiles of three attenuation coefficients with standard deviations calculated from the data in 2014. Because the KSS meteor radar has a large pulse repetition frequency (PRF), the inter-pulse period is much shorter than decay times of most observed underdense meteor trails. Hence, the pulse repetition rate factor (blue filled triangle) should be negligible in the meteor signal attenuation throughout all the altitude region and the net attenuation of meteor echo is dominated by $\alpha_r$ and $\alpha_V$ as depicted in Figure 4. The $\alpha_r$, in particular,

dramatically decreases as the initial radius ($r_0$) increases with height. This indicates that the amplitude of radar signals scattered from meteor trails is severely declined at higher altitude above about 95 km. As for the finite velocity factor, $\alpha_V$, since it is basically related to the background atmospheric state, the height variation of $\alpha_V$ remarkably coincides with that of meteor decay times, which steadily decreases with height because of the exponential decrease of the background pressure within about 82-97 km altitude range (Singer et al., 2008; Kim et al., 2010). As shown in Figure 4, the MHC generated by $\alpha_r$

and $\alpha_V$ reaches maximum (i.e., minimum attenuation coefficients) at about 100 km and this altitude is known to be a typical cutoff height for 30 MHz meteor radar observation, representing the limitation height of the observation (Olsson-Steel and Elford, 1987; Thomas et al., 1988). Because of this MHC, signals backscattered from meteor trails are significantly attenuated at higher altitudes, which causes far worse correlations between the height distribution of meteor echoes and the background atmospheric temperatures as shown in Figure 3.


As shown in Figure 4, the MHC for the KSS meteor radar is primarily controlled by the initial radius and finite velocity factors. If we assume that the distribution of meteor speed does not vary much over the 5-year observation period, the two major attenuation processes should mainly be affected by the background atmospheric density. Since the molecular mean free path is inversely proportional to the atmospheric density it is more intuitive to describe the relation between the

background atmosphere and the initial radius. Figure 5 illustrates the height distribution of meteor echoes recorded on a single day in 2016 and the height profile of the molecular mean free path calculated from the MLS pressure measurement. Note that the number of meteor echoes observed at a given height bin above the MPH more rapidly decreases with height than below. Jones and Campbell-Brown (2005) showed that the initial radius of meteor trails is about 1-2 m at the altitude of 95 to 100 km for a meteoroid falling with a speed of 40 km/s and they deduced a relationship between meteor speed $V$ and

the initial radius $r_i$: $r_i \sim V^{-0.2}$. The molecular mean free path is approximately one-third of the initial radius (Manning, 1958). When the MHC is most effective at around 97 km altitude (see Figure 5), the mean free path is about a few tens of centimeters with the initial radius of about 2~3 m, which corresponds to approximately a quarter of wavelength of the KSS meteor radar (9.03 m). This indicates that the MHC occurs within a fixed range of mean free path as shown in previous studies (Pellinen-Wannberg and Wannberg, 1994; Westman et al., 2004); in other words, it occurs at a certain atmospheric





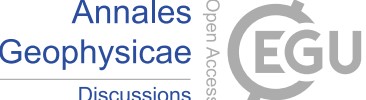

state. For the KSS meteor radar, the MHC mostly occurs at around 97 km altitude, which exists way above the MPH as shown in both Figure 4 and Figure 5. Consequently, it can be concluded that the MHC affecting meteor height distribution above the MPH is mainly controlled by the background atmospheric condition and in turn, this provides an essential validation of the temperature estimation from the FWHM.

## 5   Conclusions

In this study, the temperature estimation procedure from the FWHM is reevaluated by verifying the temporal invariance of the proportionality constant between the FWHM and mesospheric temperature over the entire observation period of 2012-2016. Their linear relationship with a proportionality constant is experimentally demonstrated from the SABER temperature and meteor radar observations in the 5-year observation period. The slope of the SABER temperature and FWHM is more consistent with theoretically derived proportionality constant than those from the MLS temperature in Lee et al. (2016). Compared to the MLS data, much better vertical resolution of the SABER temperature enabled us to find that the mesospheric temperature estimated from the FWHM represent the temperature at around $87\pm2$ km altitude, which is slightly lower than the meteor peak height by about 2-3 km. The lower representative altitude of the estimated temperature results from the asymmetric meteor echo distribution, being much lower meteor detection rates above the MPH, which is caused by the meteor echo height ceiling effect (MHC). Since the MHC well reflects the background atmospheric state, the FWHM derived from the KSS meteor radar can be used to estimate a mesospheric temperature accurately.

### Data availability

The TIMED/SABER data are available from http://saber.gats-inc.com/. The Aura/MLS data can be accessed at http://disc.sci.gsfc.nasa.gov/Aura/data-holdings/MLS. The King Sejong meteor radar data are available from the Korea Polar Research Institute upon request.

### Acknowledgements

This study was supported by the grant PE18020 from the Korea Polar Research Institute. The authors would like to thank the TIMED SABER team for providing the kinetic temperature and geopotential height (version 2.0) data. The geopotential height and pressure data from the Aura MLS team are also gratefully acknowledged.

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

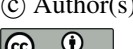



**Table 1 Slope values and correlation coefficients exhibiting a linear relationship between the SABER temperature and the FWHM from the meteor radar at KSS from 2012 to 2016.**

| Year | Number of data | Slope | $\dfrac{g}{R}\left[\ln\left(\dfrac{P_1}{P_2}\right)\right]^{-1}$ | Correlation coefficient |
|---|---|---|---|---|
| **2012** | 112 | $16.56 \pm 0.51$ | 16.17 | 0.95 |
| **2013** | 105 | $16.77 \pm 0.57$ | 16.29 | 0.95 |
| **2014** | 109 | $16.90 \pm 0.56$ | 16.29 | 0.94 |
| **2015** | 108 | $16.62 \pm 0.64$ | 16.09 | 0.94 |
| **2016** | 109 | $16.54 \pm 0.56$ | 16.31 | 0.94 |
| **2012–2016** | 543 | $16.68 \pm 0.26$ | 16.28 | 0.93 |





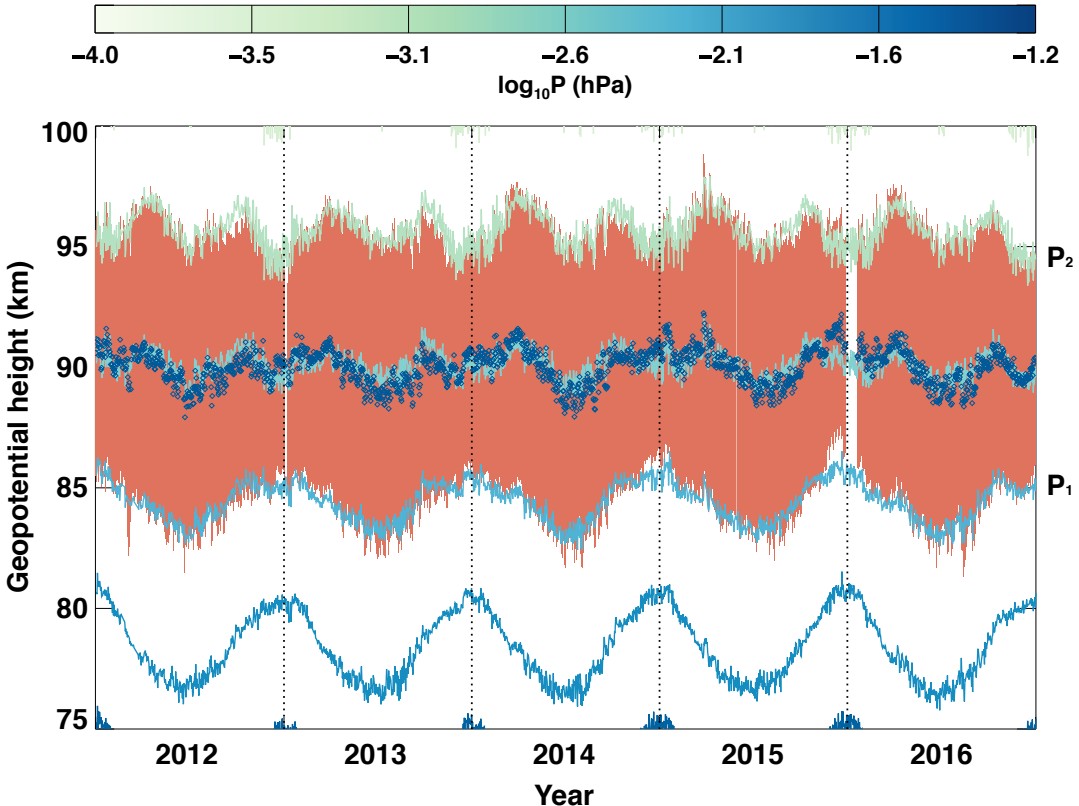

Figure 1: Temporal evolution of constant pressure surfaces of the neutral atmosphere from Aura MLS (both filled and line contours) and meteor peak detection heights (blue open diamond) with full width at half maximum (FWHM) of meteor height distribution (red shaded area) from meteor radar observations at King Sejong Station, Antarctica in 2012-2016. Two constant atmospheric pressure ($P_1$, $P_2$) levels being strongly correlated with the FWHM are also presented.





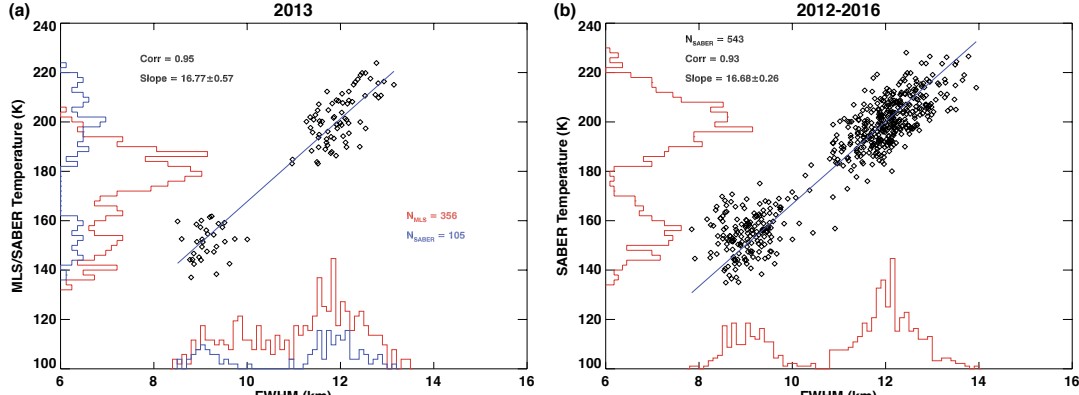

**Figure 2: Scatter plots of the daily FWHM of the meteor height distribution versus the average value of the SABER temperatures near mesopause region at King Sejong Station in (a) 2013 and (b) recent 5 years from 2012-2016. The blue solid line depicts the linear regression. The histograms of the two independent temperature measurements from the SABER (blue) and MLS (red) and**

5   **FWHM data are also presented to show the number of data used in the linear least squares.**





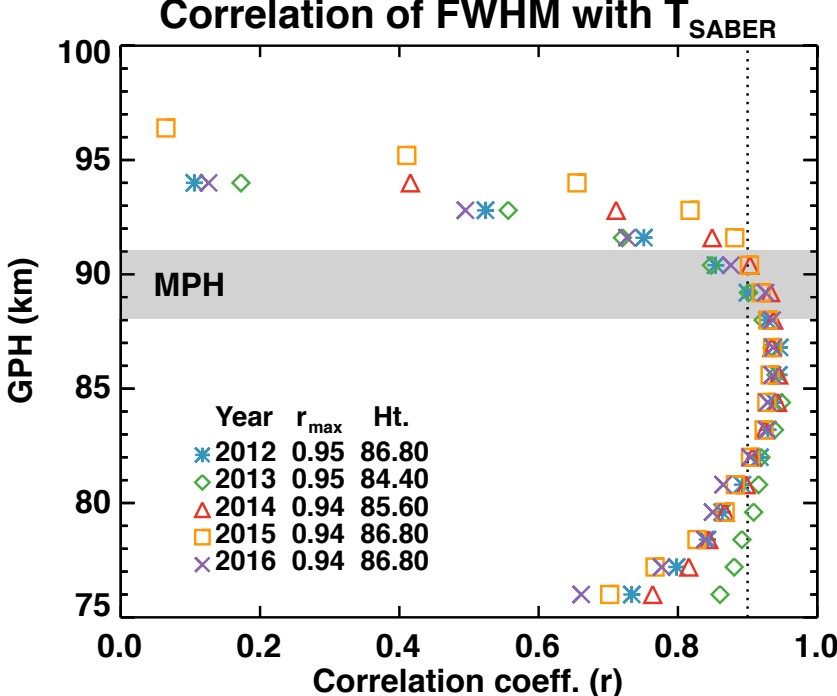

**Figure 3:** **The height profile of correlation coefficient of the FWHM and SABER temperatures in 2012-2016. The height information of the maximum correlation coefficient and its value in each year are also summarized. The dotted vertical line indicates a correlation coefficient of 0.9 and the gray shaded box denotes the height range of the MPH variation during the**

5 **observation period.**



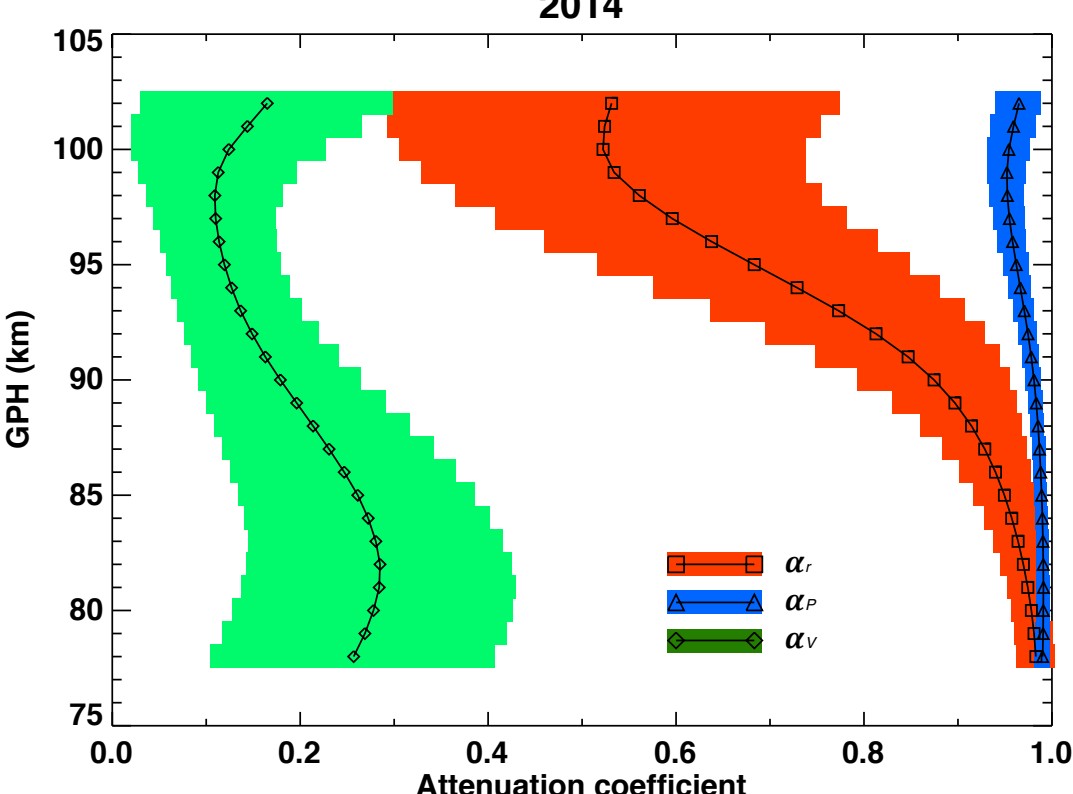

Figure 4: The height variation of yearly mean three attenuation coefficients and their one standard deviations (color-filled horizontal bars) calculated from the KSS meteor radar observations in 2014.

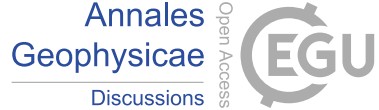



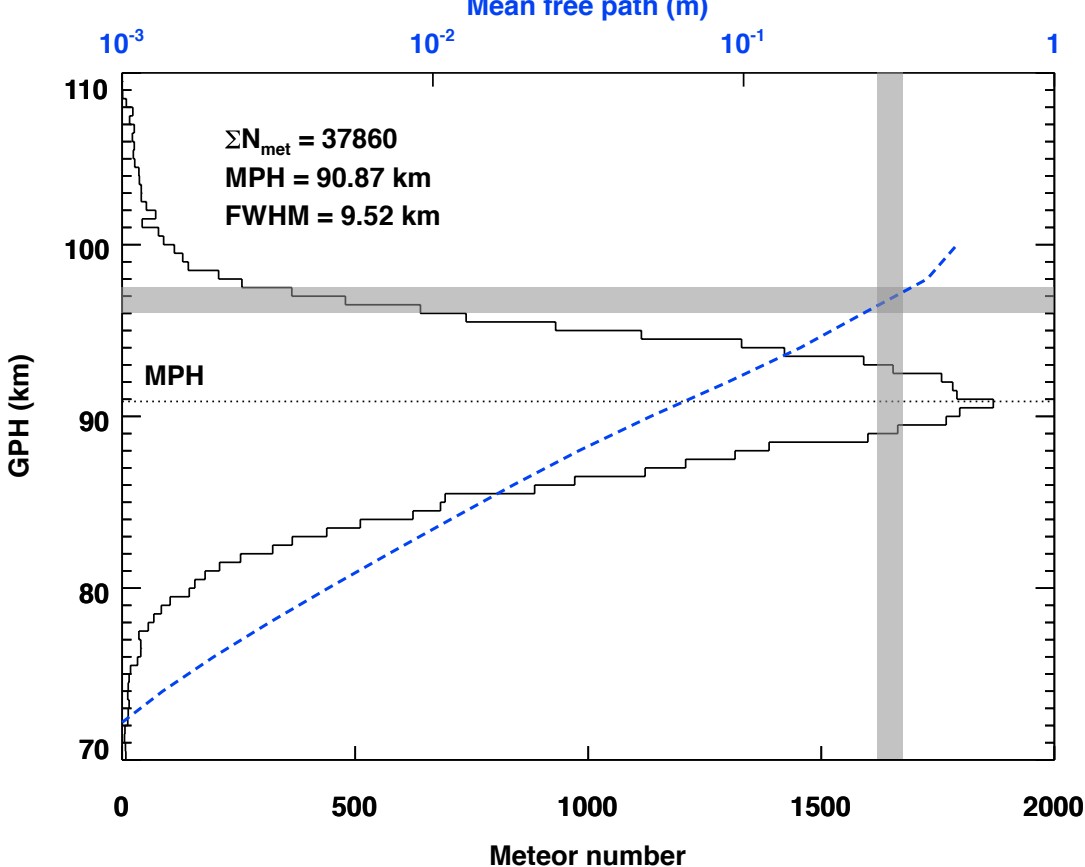

**Figure 5: The histogram of a meteor height distribution observed by the KSS meteor radar on a single day in 2016 using a 500 m bin. The blue dashed line presents the mean free path of the background atmosphere calculated from the MLS observation. The gray-colored horizontal bar indicates the height layer where rapid decrease in meteor detection rate due to the meteor echo height ceiling appears. The typical range of molecular mean free path that activates meteor echo height ceiling due to the initial radius and finite velocity factors is depicted by a gray-colored vertical bar.**