# Peer review of "Meteor echo height ceiling effect and the mesospheric temperature estimation from meteor radar observation"

_Annales Geophysicae, 2018_

## Referee Comment (RC1) · Anonymous Referee #1 · 22 May 2018

Comments: Temperature is essential information of the mesosphere. In this work the authors report the evaluation of the estimation of mesopause temperature from meteor radar echo height distribution in terms of observations from satellite observations and a meteor radar in Antarctic region. This is the extension of their previous investigation (Lee et al., GRL, 2016) with update of temperature from SABER and check the effect of meteor echo ceiling (MHC) effect on the temperature estimation. The investigation is of significant value, however, the work needs major modifications as points raised below.

Detailed comments/suggestions: 1. Page 1, Lines 19-22: The sentence should be

[Figure]

moved to the first part of paragraph. 2. Page 2, Liu et al. (Liu, L., H. Liu, H. Le, Y. Chen, Y.-Y. Sun, B. Ning, L. Hu, W. Wan, N. Li, and J. Xiong (2017), Mesospheric temperatures estimated from the meteor radar observations at Mohe, China, J. Geophys. Res. Space Physics, 122, 2249–2259, doi:10.1002/2016JA023776.) should be cited (reasons will be stated below.). 3. Page 2, Lines 10-11: Since the MHC effect, how to describe the height distribution now because the normal distribution be fail? I am curious at that they still use Guassian function to fit the distribution (Line 21, Page 3) if the MHC effect is important. 4. Page 2, Line 8: "invariance"must be deleted, because it is not so as this work presents. 5. Page 3, Lines 1-2: Since there are so limited observations from SABER over the station (the authors can check the local time coverage of SABER), how can they obtain information of geopotential height at times without SABER passes. 6. As Figure shown below for example,

the authors should be stated clearly step by step in the revised manuscript how to obtain the layer mean temperature from SABER. As there are waves in the temperature profile, how to take them into account to get the background profile? 7. More important, the SABER temperature lacks local time coverage, how to obtain daily mean temperature. If it fails to do so, how to reach the statement as given in Page 2, Lines 8-9. 8. Page 3, Lines 15-16, describe the daily profile number of SABER available over the station. 9. Page 4, Lines 20-21: It must be deleted, because Equation (1) is not valid under this case. In other words, the authors should be realized that there are assumptions being made. 10. Page 4, Lines 24-25: It should be removed as reason being given in the above and also in the Table. 11. Page 4, Lines 25-31: Words are required to tell how to get such result. 12. Page 5, Lines 3-5: no ideal local time coverage is reached for the SABER observations, how to get FWHM with geopotential height information from SABER and layered mean temperature? Figure is welcome to show it. 13. Page 5, Lines 20-23: the statement is invalid, because geopotential height of each echo should be given and the ratio of layer mean temperature to FWHM be given. 14. Page 5, Lines 30-32: It is not the same in the height range as FWHM covered. If the statement here is true, what is usefulness of Equations (1)-(3). They

are no the same now. Further, how to understand the result presented in Figure 3. I now strongly feel the authors make the layer mean temperature over FWHM and temperature at specific height confusing (although they may mean the temperature within 2.4 km). 15. Page 8, Line12: As stated above, it is misleading now. Further the statement in Page 1, Lines 11-13. 16. Figure 2: the vertical axis of left panel listed MLS, no points in the panel. 17. Figure 3: SABER temperature? Layer mean temperature over FWHM? 18. This work and Lee et al. is done with TEMPERATURE = C times FWHM, while Liu et al. [2017] adopts TEMPERATURE = C times FWHM + A. Liu et al. introduces another term A to fit the relationship between TEMPERATURE and FWHM. Further, Table 1 shows the coefficient, or C, is changing or different in years separately or together, and differs from those in column 4. At last, the authors need clarify what temperature from SABER used, layer mean temperature over FWHM range, or temperature within 2,4 km.

Please also note the supplement to this comment:
https://www.ann-geophys-discuss.net/angeo-2018-32/angeo-2018-32-RC1-supplement.pdf

---

## Referee Comment (RC2) · Anonymous Referee #2 · 29 Jun 2018

This paper clearly presents an evaluation of a method for estimating atmospheric temperature near the mesopause using the heights of meteor radar detections. As such, the content is of scientific interest and worthy of publication. The writing is clear with a few small grammatical errors that will be easy to correct. The figures are clearly presented and are integrated well with the text.

There are, however, some problems with missing references and poorly described processes that are not fully justified in the text. It is my recommendation that the paper be published following minor revisions.

Overall, it should be noted that while the authors are inferring an estimate of MLT

temperature from the width of the meteor radar detection zone, the most directly related parameter is the density scale height. A discussion of the role of scale height on the vertical extent of meteor trails is curiously absent from the manuscript. This was first discussed by Eshleman, 1957 and was investigated in detail in Younger's publicly available 2011 PhD thesis, which the lead author is familiar with.

General: The authors neglect the significant effect that meteoroid velocities have on determining the FWHM of the meteor height distribution. Faster meteors will have a smaller FWHM and are more susceptible to high-altitude cutoff. Furthermore, the relative numbers of different velocity meteoroids changes with time of day and season for a fixed observation location. Thus, the authors should calculate FWHM for a number of velocity bins and construct a fitted value for a single representative velocity, say, 30-35 km/s.

General: The asymmetry of the meteor detection height distribution is due primarily to the high-altitude cutoff. What is the effect of using the standard deviation of heights calculated separately above and below MPH?

Page 1, line 18-19: Here and throughout the paper, the authors state that they are measuring the mesopause temperature, but what is actually being estimated is a temperature near the mesopause. The height of the mesopause varies substantially more than the meteor peak height for which the authors state that their estimates are representative of.

Page 1, line 18-25: The authors should include some references to general meteor radar operation, such as McKinley, 1961, Ceplecha et all, 1998, or Holdsworth et al., 2004 (Radio Science). Furthermore, a discussion of meteor radar temperatures is incomplete without reference to Tsutsumi et al., 1994 (Radio Science) and Hocking, 1999.

Page 1, line 28-30: The authors fail to acknowledge the theoretical foundation of Eshleman, 1957, which provides the basis for their link between the height range of detected

meteors and density scale height, and thus approximate temperature.

Page 2, line 5-6: The authors should cite a paper describing the meteor radar response function, such as Cervera and Reid, 2004 or just the review paper of Ceplecha et al., 1998.

Page 2, line 16-22: For a description of what is now a standard design for meteor radars, the authors should include a reference to Jones et al., 1998 for basic concept and Holdsworth et al., 2004 (Radio Science) for the detection and analysis software used by the King Sejong MR.

Page 2, line 29: When the authors say that they limit phase error to less than six degrees, do they mean for each of the receiver channels, individual antenna pair combinations, or the array mean?

Page 3, line 27: It should be noted that atmospheric density is the determining factor in meteoroid ablation. Pressure is really only relevant in a discussion of diffusion of the meteor trail after formation.

Page 3, line 25-29: A discussion of meteoroid ablation should include a relevant reference, such as Love and Brownlee, 1999 or Rogers et al., 2005.

Page 4, line 1-10: It should be noted that this formulation is only valid for an isothermal atmosphere. This is implied later via the use of <T>, but it should be stated in the derivation. I would like to see how the FWHM compares with the density scale height, which includes a temperature gradient term.

Page 4, general: The authors' derivation and method depends on meteor detections starting and ending at two well defined pressures, P1 and P2, but they do not state why this assumption is valid. Furthermore, they provide no concrete values for P1 and P2 as used in this study and do not provide information on where they obtained theses values, although perhaps the reader is meant to infer that SABER values were used? At the very least, the authors should supply the values and uncertainties.
Page 5, line 9-12: It is worth noting that 92 km is around (and sometimes past) the upper limit of reliable measurements by the MLS instrument. As such, the vertical resolution is less important than the accuracy of values extrapolated from MLS data.

Page 5, line 17: The authors are comparing a "theoretical" prediction based on C in equation 3, but C itself is derived from experimental observations for the individual radar system. This seems like circular reasoning.

Page 5, line 26: The authors need to provide more detail than "seems plausible". It would be helpful to compare <T> obtained from their method with an average of SABER values, weighted by the distribution of meteor detections. Given the asymmetry of the meteor height distribution, would this result in a value of <T> corresponding to the lower than MPH maximum correlation height in figure 3?

Page 6, line 11: Needs reference.

Page 6, line 13-14: This statement should, at the very least, cite Jones, 1995.

Page 6, line 16-17: The destructive interference of backscatter from off-axis portions of the trail is described in detail in Younger, 2008.

Page 6, line 26: It is not just the reduced electron volume density responsible for reduced backscatter from trails with large initial radii. Backscatter from cylindrically symmetric distributions experiences significant destructive interference past the first maximum of the Bessel function in the backscatter amplitude integral (see e.g. McKinley, 1961 eq. 8-22 or Younger, 2008 figure 2).

Line 32-33: The precision of the FWHM is a purely statistical quantity determined primarily by the height accuracy of the radar and number of meteors detected. While attenuation terms do determine the behaviour of the high-altitude cutoff in detectability, it does not make sense to invoke attenuation terms in a discussion of the precision of the FWHM term.

Page 8, line 2-4: I fail to see how a demonstration of established meteor radar attenuation theory validates the authors' temperature estimation technique. The method is validated by correlation with independent measurement techniques. An assessment of attenuation coefficients is valuable for describing the shape of the meteor detection height distribution, but does not validate the method.

Figure 2: Label text in the plot area is too small to be legible.

Figure 4: This figure would be improved if the authors also showed the cumulative attenuation coefficient (product of all 3).

[Figure]

---

## Author Response (AR1)

Dear Referee #1
We're grateful for your comments and here are our responses to your comments. Our responses to each comment are written with blue-colored texts.

Page 1, Lines 19-22: The sentence should be moved to the first part of paragraph. 2.

- Thanks for your comment, but we decided to put that sentence as it is after careful deliberation. Because we need to provide a background information on how meteor trails to be produced in the introduction as several previous studies did [Lee et al., 2013; Younger et al., 2014].

  References
  1. Lee, C. S., Younger, J. P., Reid, I. M., Kim, Y. H. and Kim, J. H.: The effect of recombination and attachment on meteor radar diffusion coefficient profiles, J Geophys Res-Atmos, 118(7), 3037–3043, doi:10.1002/jgrd.50315, 2013.
  2. Younger, J. P. C. S. L. I. M. R. R. A. V. Y. H. K. A. D. J. M.: The effects of deionization processes on meteor radar diffusion coefficients below 90 km,, 1–17, doi:10.1002/(ISSN)2169-8996, 2014.

Page 2, Liu et al. (Liu, L., H. Liu, H. Le, Y. Chen, Y.-Y. Sun, B. Ning, L. Hu, W. Wan, N. Li, and J. Xiong (2017), Mesospheric temperatures estimated from the meteor radar observations at Mohe, China, J. Geophys. Res. Space Physics, 122, 2249–2259, doi:10.1002/2016JA023776.) should be cited (reasons will be stated below.).

- We'll added that paper as reference. Thanks.

3. Page 2, Lines 10-11: Since the MHC effect, how to describe the height distribution now because the normal distribution be fail? I am curious at that they still use Guassian function to fit the distribution (Line 21, Page 3) if the MHC effect is important.

- As you pointed out, the meteor radar observation at high altitude is affected by MHC effect and this makes asymmetry in the meteor height distribution as shown in the figure below. However, the extent of the asymmetry is not very severe and the Gaussian function is still the suitable model to determine the best FWHM values to be compared to the SABER temperature.

[Figure]

[Lee et al., 2016]

4. Page 2, Line 8: "invariance" must be deleted, because it is not so as this work presents.

- This study presents the linear relationship between SABER temperature and FWHM based on the fact that the meteor height distribution is primarily controlled by the background atmospheric pressures as shown in Figure 1. The proportionality constant between temperature and FWHM is defined to be a constant as in the equation (3), which was demonstrated from the observational data within measurement errors over the 5-year period as shown in Table 1 in the manuscript. And this is the key idea of the temperature estimation procedure using the observed FWHM, instead of using diffusion coefficient. Once it is determined from the independent measurements such as SABER in our study, the daily mean temperature can be estimated from the meteor radar observed FWHM alone without any additional information.

5. Page 3, Lines 1-2: Since there are so limited observations from SABER over the station (the authors can check the local time coverage of SABER), how can they obtain information of geopotential height at times without SABER passes.

- We agree that the SABER only scan two local times over one local position a day as it has a sun synchronous orbit but, we don't need SABER data to get geopotential height from the meteor echo data. There is a simple equation to convert between the geometric height (h) and geopotential height (hg) as follows,

$$hg=h*(r/r-h),$$

where r is the earth's radius. Based on this formula, all the geometric heights of meteor echo can be easily converted to geopotential height without SABER data.

6. As Figure shown below for example,

[Figure]

the authors should be stated clearly step by step in the revised manuscript how to obtain the layer mean temperature from SABER. As there are waves in the temperature profile, how to take them into account to get the background profile?

- Since the accumulated meteor height distribution during a day only provides one FWHM, it is more natural that the FWHM can reflect the daily mean temperature not the temperature at the moment. The layer mean temperature corresponds to the red solid line (of course there is height-bin dependence) and the red solid line still shows mean temperature information even if there is a wave structure in the profile. Daily mean temperature can be obtained by averaging the at least two individual temperature profiles (your figure is a single temperature profile at 13.81 UT) and wave structures is more likely getting weaker or even smoothed out in the average procedure.

7. More important, the SABER temperature lacks local time coverage, how to obtain daily mean temperature. If it fails to do so, how to reach the statement as given in Page 2, Lines 8-9.

- Since the SABER only covers two separated local times (day and night for each) over any geographic locations, we calculated mean temperature profile from the SABER temperature data recorded on a single day of year. Several previous studies [Meek et al., 2013; Holmen et al., 2016; Yi et al., 2016] used spatial grid to limit MLS or SABER temperature to the specific location for direct comparison with local meteor radar data and we also did in the same way. Determining the spatial grid for data selection is a tradeoff between number of available satellite data and accurate comparison with the local ground-based

measurement.

References
1. Meek, C. E., Manson, A. H., Hocking, W. K. and Drummond, J. R.: Eureka, 80° N, SKiYMET meteor radar temperatures compared with Aura MLS values, Annales Geophysicae, 31(7), 1267–1277, doi:10.5194/angeo-31-1267-2013, 2013.
2. Holmen, S. E., Hall, C. M. and Tsutsumi, M.: Neutral atmosphere temperature trends and variability at 90 km, 70°N, 19°E, 2003–2014, Atmos. Chem. Phys., 16(12), 7853–7866, doi:10.5194/acp-16-7853-2016, 2016.
3. Yi, W., Xue, X., Chen, J., Dou, X., Chen, T. and Li, N.: Estimation of mesopause temperatures at low latitudes using the Kunming meteor radar, Radio Sci., 51(3), 130–141, doi:10.1002/2015RS005722, 2016.

8. Page 3, Lines 15-16, describe the daily profile number of SABER available over the station.

- When we limit SABER data to the distance of less than 500 km from the location of KSS, 3-4 profiles are available on average.

9. Page 4, Lines 20-21: It must be deleted, because Equation (1) is not valid under this case. In other words, the authors should be realized that there are assumptions being made.

- The thermodynamic state of the atmosphere at any point is determined by pressure, temperature and density. These variables are related to each other by the ideal gas law. The hydrostatic balance provides an excellent approximation for the vertical dependence of the pressure field in the real atmosphere [Andrew et al., 1987; Holton, 2004; North et al., 2014]. Of course the real atmosphere is different from its ideal state but they work very well. Below references obviously show that ideal gas law and hydrostatic equation can be used to describe atmospheric physics. It would be appreciated if you provide more appropriate equations better describing the FWHM and atmospheric pressure field than equation (1).

References
1. Andrew, D. G., Holton, J. R., Leovy, C. B., Middle Atmospheric Dynamics, Academic Press. 1987.
2. Holton, J. R., An introduction to dynamic meteorology, vol. 88, Academic Press, 2004.
3. North, G. R., Pyle, J. A. and Zhang, F.: Encyclopedia of Atmospheric Sciences, Elsevier. 2014.

10. Page 4, Lines 24-25: It should be removed as reason being given in the above and also in the Table.

- As we already mentioned in previous response to comment 4, time-invariance of proportionality constant is a fundamental idea to make the FWHM estimate background atmospheric temperature. Otherwise whenever we determine the atmospheric temperature from the FWHM, we need SABER or MLS temperatures to conduct linear regression procedure. This study wants to tell that the temperature can be estimated from the FWHM alone without any further information. The proportionality constant in the table has its own standard error due to uncertainties in FWHM and SABER temperature measurements, please note that the constant does not change within a given standard errors during the entire periods.

11. Page 4, Lines 25-31: Words are required to tell how to get such result.

- Firstly, we try to find the two height layers where the envelopes of the FWHM meet (please refer to figure 1 in the manuscript) and SABER pressure values at those two height layers can be found every day. Once two pressure values over the entire observational period are recorded, we calculate mean value of two pressures (P1, P2) and they can be used to obtain the proportionality constant from $C = \frac{g}{R}\left[\ln\left(\frac{P_1}{P_2}\right)\right]^{-1}$

12. Page 5, Lines 3-5: no ideal local time coverage is reached for the SABER observations, how to get FWHM with geopotential height information from SABER and layered mean temperature? Figure is welcome to show it.

- We already mentioned how to get geopotential height without SABER data in our response to comment 5. All the geometric height of meteor echo data can be converted to geopotential height using a simple formula.

13. Page 5, Lines 20-23: the statement is invalid, because geopotential height of each echo should be given and the ratio of layer mean temperature to FWHM be given.

- From the simple relation in our response to comment 5 between the geometric and geopotential height we already obtained all the geopotential height from the meteor echo data. Based on the linear relationship between the FWHM and the temperature, T = C*FWHM, we can calculate the daily mean temperature directly using FWHM alone. Lee et al., (2016) already showed that FWHM can provide better temperature estimation with lower uncertainties than meteor decay times.

  Reference
  Lee, C., Kim, J. -H., Jee, G., Lee, W., Song, I. S. and Kim, Y. H., New method of estimating temperatures near the mesopause

region using meteor radar observations, Geophys. Res. Lett., 43(2), 10, doi:10.1002/2016GL071082, 2016.

14. Page 5, Lines 30-32: It is not the same in the height range as FWHM covered. If the statement here is true, what is usefulness of Equations (1)-(3).
They are no the same now.   Further, how to understand the result presented in Figure 3. I now strongly feel the authors make the layer mean temperature over FWHM and temperature at specific height confusing (although they may mean the temperature within 2.4 km).

  - According to your comment, we'll add more description in data analysis part for better understanding about the layer mean temperature. All the equations are essential to approve the linear relationship between the FWHM and background temperature based on the fact that the FWHM corresponds to the height difference between two fixed atmospheric pressures as shown in figure 1 in the manuscript. When we compare the proportionality constant from the least-squares fitting with one from the equation $C = \frac{g}{R}\left[\ln\left(\frac{P_1}{P_2}\right)\right]^{-1}$, the height difference between P1 and P2 have to be identical to the FWHM, and we have done in that way as we explained in our response to comment 11. Since the SABER has a better vertical resolution than MLS, we can find representative height where the FWHM can estimate the temperature by using finer height bin size. If we used height bin size comparable to the FWHM instead, we have to assume that FWHM estimate atmospheric temperature near the meteor peak height as Lee et al., (2016) did using MLS.

  - In this study, we started with calculation the layer mean temperature having a height bin of 2.4 km and further analyses were conducted.

15.   Page 8, Line12:   As stated above, it is misleading now.   Further the statement in Page 1, Lines 11-13.

  - We're very sorry to say this but, we don't understand where misleading part is. Page 8 line12 and page1 line11-13 tell the exactly same feature that the FWHMs have the best correlation with the temperature at around 87 km, which is a little lower than the meteor peak height (~90 km). We also pointed out that rapid decrease in correlation coefficients above MPH is caused by the MHC effect. You can easily find above description in figure 3 in the manuscript.

16.   Figure 2:   the vertical axis of left panel listed MLS, no points in the panel.

  - Yes. We admit that MLS data was not used in this study, but we just want to show that SABER has less number of available data compared to MLS due to its limited geometrical coverage for high-latitude (> 52 degrees) regions as

described in the manuscript. This can be used to explain higher fitting error of proportionality constant in least-squares fitting procedure.

17.   Figure 3:   SABER temperature?   Layer mean temperature over FWHM?

-   As we described in our response to comment 14, SABER temperature data were interpolated every 1.2 km first and the layer mean temperatures were obtained within 2.4 km height bin. This means the height layer for mean temperature is overlapped by 50 % (height-bin=2.4 km height step=1.2 km). You can find there are 10 data points in 12 km height region in the figure below.

[Figure]

18.   This work and Lee et al. [2016] is done with TEMPERATURE= C times FWHM, while Liu et al.   [2017] adopts TEMPERATURE = C times FWHM +A. Liu et al. introduces another term A to fit the relationship between TEMPERATURE and FWHM. Further,   Table 1 shows the coefficient,   or C, is changing or different in years separately or together, and differs from those in column 4.   At last, the authors need clarify what temperature from SABER used, layer mean temperature over FWHM range, or temperature within 2,4 km.

-   The linear relationship between the temperature and the FWHM is derived from the basic equations (ideal gas law and hydrostatic equation) and we also showed that the FWHM closely follows background atmospheric pressures (P1, P2) from independent observations. In this study, we clearly showed the physical meaning of "T=C*FWHM" and C should be considered as the constant over 5-year observational period under a given uncertainty.

-   When we used "T=C*FWHM+A" form as Liu et al. (2017) did to define the relationship between FWHM and the SABER temperature, both C and A

dramatically changes for each year and they are unpredictable as summarized in the table below. What if we have to estimate the mesospheric temperature using "T=C*FWHM+A" in 2016 or 2017 ? We can easily expect that independent temperature measurement from the MLS or SABER is necessary to find new C and A in a given period.

| Year | C | A |
|------|------|------|
| 2012 | 15.11 ± 0.03 | 16.52 ± 0.35 |
| 2013 | 17.23 ± 0.04 | -5.11 ± 0.45 |
| 2014 | 13.67 ± 0.03 | 36.71 ± 0.31 |
| 2015 | 11.82± 0.03 | 55.08. ± 0.30 |

Dear Referee #2

We greatly appreciate your constructive comment for thoughtful evaluations of the manuscript and helpful suggestions for its improvement. We did our best to response to all your comments. Author's responses were written in blue text below every referee's comment.

This paper clearly presents an evaluation of a method for estimating atmospheric temperature near the mesopause using the heights of meteor radar detections. As such, the content is of scientific interest and worthy of publication. The writing is clear with a few small grammatical errors that will be easy to correct. The figures are clearly presented and are integrated well with the text.

There are, however, some problems with missing references and poorly described processes that are not fully justified in the text. It is my recommendation that the paper be published following minor revisions.

Overall, it should be noted that while the authors are inferring an estimate of MLT temperature from the width of the meteor radar detection zone, the most directly related parameter is the density scale height. A discussion of the role of scale height on the vertical extent of meteor trails is curiously absent from the manuscript. This was first discussed by Eshleman, 1957 and was investigated in detail in Younger's publicly available 2011 PhD thesis, which the lead author is familiar with.

- We agree that it is very important to mention about density scale height. From the ideal gas law and hydrostatic equation as shown from Eq(1) to Eq(3), scale height (mg/kT) should correspond to ln(P1/P2)/FWHM because we can readily derive the simple formula from ideal gas law and hydrostatic eqution as below,

$$\ln\frac{P_1}{P_2} = \frac{mg}{kT}(Z_2 - Z_1)$$

As described in the manuscript, $Z_2 - Z_1$ is identical to the FWHM.

- Since we defined layer mean temperature <T>, the height region of interest in this study can be considered isothermal. In the manuscript, the ideal gas law was written as P=$\rho$RT instead of P=nkT. According to your comment, we added description of the scale height in temperature estimation from the height width of meteor distribution with relevant references. (Eshleman, 1957; Younger, 2011).

General: The authors neglect the significant effect that meteoroid velocities have on determining the FWHM of the meteor height distribution. Faster meteors will have a smaller FWHM and are more susceptible to high-altitude cutoff. Furthermore, the relative numbers of different velocity meteoroids changes with time of day and season for a fixed observation location. Thus, the authors should calculate FWHM for a number of velocity bins and construct a fitted value for a single representative velocity, say, 30-35 km/s.

- We totally agree with your comment and we'll calculate FWHM from representative meteor velocity like 30-35 km/s after we check the dependence of the FWHM on meteoroid speed.

General: The asymmetry of the meteor detection height distribution is due primarily to the high-altitude cutoff. What is the effect of using the standard deviation of heights calculated separately above and below MPH?

- Although we have not calculated standard deviation of heights separately above and below MPH, we obtained separate height widths from meteor detection region below and above MPH by independent Gaussian curves to height regions. That means the FWHM can be expressed as sum of half widths of two fitted curves. Unfortunately, the FWHM from two separated height widths gave us worse temperature estimation compared to the FWHM from a unified Gaussian fitting curve or even to traditional meteor decay method. As the figure 1 in Lee et al. (2016) clearly shows, the magnitude of asymmetry in meteor height distribution is very small.

Page 1, line 18-19: Here and throughout the paper, the authors state that they are measuring the mesopause temperature, but what is actually being estimated is a temperature near the mesopause. The height of the mesopause varies substantially more than the meteor peak height for which the authors state that their estimates are representative of.

- According to your comment, we changed "mesopause temperature" to " temperature near the mesopause". Thanks.

Page 1, line 18-25: The authors should include some references to general meteor radar operation, such as McKinley, 1961, Ceplecha et all, 1998, or Holdsworth et al., 2004 (Radio Science). Furthermore, a discussion of meteor radar temperatures is incomplete without reference to Tsutsumi et al., 1994 (Radio Science) and Hocking, 1999.

- Following your comment, we added all references in radar operation and meteor radar temperature description. Thanks.

Page 1, line 28-30: The authors fail to acknowledge the theoretical foundation of Eshleman, 1957, which provides the basis for their link between the height range of detected meteors and density scale height, and thus approximate temperature.

- We added statement "Eshleman [1957] provided a theoretical basis for the relationship between the atmospheric density scale height and the height range of detected meteor echoes. This relationship was developed by showing that the width of the height distribution of detected meteors is a nearly linear function of the density scale height [Younger 2011]." based on your comment.

Page 2, line 5-6: The authors should cite a paper describing the meteor radar response function, such as Cervera and Reid, 2004 or just the review paper of Ceplecha et al., 1998.

- We cited Cervera and Reid, 2004. Thanks.

Page 2, line 16-22: For a description of what is now a standard design for meteor radars, the authors should include a reference to Jones et al., 1998 for basic concept and Holdsworth et al., 2004 (Radio Science) for the detection and analysis software used by the King Sejong MR.

- According to the comment, we cited Jones et al., 1998 for the configuration of receiver array and Holdsworth et al., 2004 for the meteor radar data analysis.

Page 2, line 29: When the authors say that they limit phase error to less than six degrees, do they mean for each of the receiver channels, individual antenna pair combinations, or the array mean?

- Phase error in the manuscript means that mean value of phase difference error for the individual antenna pair combinations. We added description to make it clear. Thanks.

Page 3, line 27: It should be noted that atmospheric density is the determining factor in meteoroid ablation. Pressure is really only relevant in a discussion of diffusion of the meteor trail after formation.

- We agree that density primarily controls meteoroid ablation and pressure is a function of density and temperature from the ideal gas law. When we compared density field derived from Aura/MLS and height width of meteor distribution (FWHM), we found that the FWHM had better correlation with the pressure than the density. Based on this, we think the height distribution of detected meteor echoes is determined by not only density but background temperature.

Page 3, line 25-29: A discussion of meteoroid ablation should include a relevant reference, such as Love and Brownlee, 1999 or Rogers et al., 2005.

- According to your comment, we added two papers as reference. Thanks.

Page 4, line 1-10: It should be noted that this formulation is only valid for an isothermal atmosphere. This is implied later via the use of <T>, but it should be stated in the derivation. I would like to see how the FWHM compares with the density scale height, which includes a temperature gradient term.

- Once the layer mean temperature, <T> is defined as eq (4), eq (3) can be readily derived by dividing eq (2) by $\int_{P_2}^{P_1} d \ln P$. From the FWHM, we can estimate averaged temperature within a layer of two pressure values and the layer mean temperature can define any kind of atmosphere even rapid varying temperature profile. As shown in figure 1, FWHMs well follow constant atmospheric pressure region (P1, P2) and this observationally supports eq (3).

Page 4, general: The authors' derivation and method depends on meteor detections starting and ending at two well defined pressures, P1 and P2, but they do not state why this assumption is valid. Furthermore, they provide no concrete values for P1 and P2 as used in this study and do not provide information on where they obtained theses values, although perhaps the reader is meant to infer that SABER values were used? At the very least, the authors should supply the values and uncertainties.

- As Lee et al., (2016) did, we assumed that meteor height distribution is mainly determined by background atmosphere from two independent observations for 5 years such as meteor height distributions from meteor radar and atmospheric pressures from Aura/MLS. To prove this assumption is correct, we used two fundamental equations (ideal gas law, hydrostatic equation) and derived hypsometric equation which obviously showed the linear relationship between the layer mean temperature and the FWHM.

- Based on your comment, we presented 5-year averaged values of P1, P2 with standard deviation calculated from SABER measurements in the manuscript and relevant histogram is added as below,

[Figure]

Page 5, line 9-12: It is worth noting that 92 km is around (and sometimes past) the upper limit of reliable measurements by the MLS instrument. As such, the vertical resolution is less important than the accuracy of values extrapolated from MLS data.

- We agree with your comment, but please note that so many previous studies evaluating temperature [Meek et al., 2013; Kozlovsky et al., 2016; Yi et al., 2016; Lee et al., 2016] and density [Younger et al., 2015; Yi et al., 2018] estimation from meteor radar used MLS temperature/pressure measurements. In this study, we try to find specific height of temperature estimated from the meteor height distribution and this is a main reason why we used SABER data instead of MLS. Because the SABER has better vertical resolution and larger altitude coverage than MLS does.

Page 5, line 17: The authors are comparing a "theoretical" prediction based on C in equation 3, but C itself is derived from experimental observations for the individual radar system. This seems like circular reasoning.

- Firstly, we calculated proportionality constant (C1) between the SABER temperature and the FWHM by least-squares method and C1 should be considered empirical value of proportionality constant. From hypsometric equation, we calculated $C2 = \frac{g}{R}\left[\ln\left(\frac{P_1}{P_2}\right)\right]^{-1}$, which corresponds to proportionality constant, C in eq (3). Although both C1 and C2 represent the proportionality constant between the temperature and the FWHM, they have been derived from independent methods. When we obtained C2 from the

hypsometric equation, realistic values of (P1, P2) are required and those pressure values were obtained from SABER measurements.

- Based on your comment, we replaced "theoretical values" by "constant in eq (3) with SABER pressure measurements".

Page 5, line 26: The authors need to provide more detail than "seems plausible". It would be helpful to compare <T> obtained from their method with an average of SABER values, weighted by the distribution of meteor detections. Given the asymmetry of the meteor height distribution, would this result in a value of <T> corresponding to the lower than MPH maximum correlation height in figure 3?

- Since the FWHM can be defined around the MPH, it is natural to assume that the temperature derived from the FWHM can represent the mean temperature at near the MPH. However, we showed that the representative height of temperature estimated from the FWHM is slightly lower than the MPH by 3-4 km in correlation analysis. We thought that the lower representative height and the asymmetry of the meteor height distribution should be caused by the meteor height ceiling (MHC) effect.

Page 6, line 11: Needs reference. Page 6, line 13-14: This statement should, at the very least, cite Jones, 1995.

- We added Thomas et al., 1988; Steel and Elford, 1991 as references. Jones, 1995 was cited as you commented in line 13-14. Thanks.

Page 6, line 16-17: The destructive interference of backscatter from off-axis portions of the trail is described in detail in Younger, 2008.

- Younger 2008 paper was added in line 16-17. Thanks.

Page 6, line 26: It is not just the reduced electron volume density responsible for reduced backscatter from trails with large initial radii. Backscatter from cylindrically symmetric distributions experiences significant destructive interference past the first maximum of the Bessel function in the backscatter amplitude integral (see e.g. McKinley, 1961 eq. 8-22 or Younger, 2008 figure 2).

- We're grateful for your comment in Bessel function dependence of backscattered signal amplitude. According to the comment, we corrected the statement as "The reduced electron density and its weighting function (zeroth-order Bessel function) oscillating positive and negative regions with a radial distance in the meteor trail…"

Line 32-33: The precision of the FWHM is a purely statistical quantity determined primarily by the height accuracy of the radar and number of meteors detected. While attenuation terms do determine the behaviour of the high-altitude cutoff in detectability, it does not make sense to invoke attenuation terms in a discussion of the precision of the FWHM term.

- we totally agree with your comment and we modified the sentence to avoid misunderstanding. Thanks. The corrected statement is as follows,

- "Although the background atmospheric pressure field primary factor to determine the FWHM, the MHC also contributes to the FWHM by reducing the detection of high altitude meteor trails."

Page 8, line 2-4: I fail to see how a demonstration of established meteor radar attenuation theory validates the authors' temperature estimation technique. The method is validated by correlation with independent measurement techniques. An assessment of attenuation coefficients is valuable for describing the shape of the meteor detection height distribution, but does not validate the method.

- As we described in the last paragraph in page 7 with figure 4 and figure 5, the MHC effect is mainly controlled by initial radius factor. From the relationship between neutral density (molecular mean free path) and initial radius, the MHC mostly occurs within a fixed mean free path supporting previous studies.

- Since the MHC produces asymmetric structure in meteor height distribution due to the high-altitude cutoff in detectability and this means that the MHC decreases the FWHM in meteor height distribution. As shown in table 1, proportionality constants (C1) from the least-squares method using SABER temperature and the FWHM tend to be slightly larger (by 1.4 ~ 3.7 %) than values (C2) from eq (3) with SABER pressure measurements. We thought that underestimated FWHM under the MHC effect provided the reason why C1s are systematically larger than C2 over the entire observation period.

- It should be noted that the MHC reduce the FWHM more effectively in winter when broader meteor height distributions (larger FWHMs) appear than summer because the upper part of FWHM in winter easily reaches cutoff height (~97 km) of MHC. This makes the empirical slope (C1) larger as shown in the figure below,

[Figure]

- In summary, although the MHC affects the absolute value of the FWHM and produces lower representative height of temperature estimation, it well reflects background atmospheric condition because it only happens at a constant atmospheric density (or mean free path).

Figure 2: Label text in the plot area is too small to be legible.

- We used bigger label text for legibility in figure 2. Thanks.

Figure 4: This figure would be improved if the authors also showed the cumulative attenuation coefficient (product of all 3).

- We added the normalized cumulative attenuation coefficient in the right hand. Thanks.

[revised manuscript text omitted]

**Cervera, M. A., and I. M. Reid (2000), Comparison of atmospheric parameters derived from meteor observations with CIRA, Radio Sci., 35(3), 833–843, doi: 10.1029/1999RS002226.**

Chilson, P. B., P. Czechowsky, and G. Schmidt (1996), A comparison of ambipolar diffusion coefficients in meteor trains using VHF radar and UV lidar, Geophys. Res. Lett., 23(20), 2745–2748, doi:10.1029/96GL02577.

**Eshleman, V. R. (1957), The Theoretical Length Distribution of Ionized Meteor Trails, J. Atmos. Terr. Phys., 10, 57-72.**

Holdsworth, D. A., R. J. Morris, D. J. Murphy, I. M. Reid, G. B. Burns, and W. J. R. French (2006), Antarctic mesospheric temperature estimation using the Davis mesosphere-stratosphere-troposphere radar, J. Geophys. Res., 111, D05108, doi:10.1029/2005JD006589.

**Jones, W. (1995), Theory of the initial radius of meteor trains, Monthly Notices of the Royal Astronomical Society, 275(3), 812–818.**

Jones, J., A. R. Webster, and W. K. Hocking (1998), An improved interferometer design for use with meteor radars, Radio Sci., 33(1), 55–65, doi: 10.1029/97RS03050.

Jones, J., and M. Campbell Brown (2005), The initial train radius of sporadic meteors, Monthly Notices of the Royal Astronomical Society, 359(3), 1131–1136, doi:10.1111/j.1365-2966.2005.08972.x.

Kim, J.-H., Y. H. Kim, C. S. Lee, and G. Jee (2010), Seasonal variation of meteor decay times observed at King Sejong Station (62.22°S, 58.78°W), Antarctica, Journal of Atmospheric and Solar-Terrestrial Physics, 72(11-12), 883–889, doi:10.1016/j.jastp.2010.05.003.

Kim, J.-H., Y. H. Kim, G. Jee, and C. Lee (2012), Mesospheric temperature estimation from meteor decay times of weak and strong meteor trails, Journal of Atmospheric and Solar-Terrestrial Physics, 89(C), 18–26, doi:10.1016/j.jastp.2012.07.003.

Lee, C., Y. H. Kim, J.-H. Kim, G. Jee, Y.-I. Won, and D. L. Wu (2013), Seasonal variation of wave activities near the mesopause region observed at King Sejong Station (62.22°S, 58.78°W), Antarctica, Journal of Atmospheric and Solar-Terrestrial Physics, 105, 30–38, doi:10.1016/j.jastp.2013.07.006.

Lee, C., J.-H. Kim, G. Jee, W. Lee, I.-S. Song, and Y. H. Kim (2016), New method of estimating temperatures near the mesopause region using meteor radar observations, Geophys. Res. Lett., 43, 10,580–10,585, doi:10.1002/2016GL071082.

Liu, L., H. Liu, H. Le, Y. Chen, Y.-Y. Sun, B. Ning, L. Hu, W. Wan, N. Li, and J. Xiong (2017), Mesospheric temperatures estimated from the meteor radar observations at Mohe, China, J. Geophys. Res. Space Physics, 122, 2249–2259, doi:10.1002/2016JA023776.

**Love, S. G. and D. E. Brownlee (1991), Heating and thermal transformation of micrometeoroids entering the earth's atmosphere. Icarus, 89, 26–43.**

Manning, L. A. (1958), The Initial Radius of Meteoric Ionization Trails, J. Geophys. Res., 63(1), 181–196, doi:10.1029/JZ063i001p00181.

McKinley, D. W. R. (1961), Meteor Science and Engineering, McGraw-Hill, New York.

Meek, C. E., A. H. Manson, W. K. Hocking, and J. R. Drummond (2013), Eureka, 80° N, SKiYMET meteor radar temperatures compared with Aura MLS values, Annales Geophysicae, 31(7), 1267–1277, doi:10.5194/angeo-31-1267-2013.

Mertens, C. J., M. G. Mlynczak, M. López-Puertas, P. P. Wintersteiner, R. H. Picard, J. R. Winick, L. L. Gordley, and J. M. Russell III (2001), Retrieval of mesospheric and lower thermospheric kinetic temperature from measurements of CO2 15-mm Earth limb emission under non-LTE conditions, Geophys. Res. Lett., 28, 1391–1394.

Mertens, C. J., et al. (2004), SABER observations of mesospheric temperatures and comparisons with falling sphere measurements taken during the 2002 summer MaCWAVE campaign, Geophys. Res. Lett., 31, L03105, doi:10.1029/2003GL018605.

Olsson-Steel, D., and W. G. Elford (1987), The true height distribution and flux of radar meteors, IN: European Regional Astronomy Meeting of the IAU, 67, 193–197.

Pellinen-Wannberg, A., and G. Wannberg (1994), Meteor observations with the European Incoherent Scatter UHF Radar, J. Geophys. Res., 99(A6), 11379–11390, doi:10.1029/94JA00274.

**Rogers, L. A., K. A. Hill, and R. L. Hawkes (2005), Mass loss due to sputtering and thermal processes in meteoroid ablation. Plan. Space Sci., 53, 1341–1354.**

Singer, W., R. Latteck, L. F. Millan, N. J. Mitchell, and J. Fiedler (2008), Radar Backscatter from Underdense Meteors and Diffusion Rates, Earth, 102(1), 403–409, doi:10.1007/s11038-007-9220-0.

Steel, D. I., and W. G. Elford (1991), The Height Distribution of Radio Meteors - Comparison of Observations at Different Frequencies on the Basis of Standard Echo Theory, J. Atmos. Terr. Phys., 53(5), 409–417, doi:10.1016/0021-9169(91)90035-6.

Thomas, R. M., P. S. Whitham, and W. G. Elford (1988), Response of high frequency radar to meteor backscatter, J. Atmos. Terr. Phys., 50(8), 703–724, doi:10.1016/0021-9169(88)90034-7.

**Tsutsumi, M., T. Tsuda, T. Nakamura, and S. Fukao (1994), Temperature fluctuations near the mesopause inferred from meteor observations with the middle and upper atmosphere radar, Radio Sci., 29(3), 599–610, doi: 10.1029/93RS03590.**

**Younger, J. P., Reid, I. M., Vincent, R. A. and Holdsworth, D. A. (2008), Modeling and observing the effect of aerosols on meteor radar measurements of the atmosphere, Geophys. Res. Lett., 35(1), L15812, doi:10.1029/2008GL033763.**

**Younger, J. P. (2011), Theory and Applications of VHF Meteor Radar Observations, Ph.D. dissertation, University of Adelaide.**

Westman, A., Wannberg, G., and A. Pellinen-Wannberg, (2004), Meteor head echo altitude distributions and the height cutoff effect studied with the EISCAT HPLA UHF and VHF radars, Ann. Geophys., 22, 1575-1584, https://doi.org/10.5194/angeo-22-1575-2004, 2004.